# New Strategies for Potential Contrast Agents' Synthons Highly Active to MRI Based on $Gd^{3+}$, $Eu^{3+}$, and $Tb^{3+}$

Carlos Guzmán [1], Rubén Soria-Martínez [2] and Julián Urresta [1,*]

1 Chemistry Department, Faculty of Natural and Exact Sciences, Universidad del Valle, Meléndez University City Street 13, #100-00 Cali, Colombia
2 Centro Brasileiro de Pesquisas Físicas-COMAN, R. Dr. Xavier Sigaud, 150-Urca, Rio de Janeiro-RJ 22290-180, Brazil
* Correspondence: julian.urresta@correounivalle.edu.co

**Featured Application: Authors are encouraged to provide a concise description of the specific application or a potential application of the work. This section is not mandatory.**

**Abstract:** The library of new smart contrast agents based on $Gd^{3+}$, $Eu^{3+}$, and $Tb^{3+}$ used as biomarkers is in continuous development due to its applications in diagnostic imaging. The search for safer and more efficient contrast agents has focused on the design of compounds that exhibit high relaxivity. Herein, we present alternative synthetic strategies for the development of theoretically high-relaxivity synthons based on lanthanides using the Solomon–Bloembergen–Morgan equations through click chemistry and direct addition. Special attention has been devoted to the analysis of the different aspects interfering with the successful acquisition of these complexes and their troubleshooting during their synthesis. Our preliminary results showed that not only the mathematical background needs to be considered, but also the synthetic strategy and the use of procedures free of metallic ions favor the total synthesis of these challenging complexes.

**Keywords:** click chemistry; gadolinium; MRI; relaxivity; contrast agent; SBM theory

## 1. Introduction

The strong need for outpatient surgery, non-invasive testing, and highly accurate techniques for the early detection of elusive pathologies such as cancer have become imperative to advancing effective theragnostic strategies. Therefore, imaging techniques such as magnetic resonance imaging (MRI) and X-rays have come to be the main diagnostic imaging methods [1]. However, these nuclear techniques are limited by their poor resolution to differentiate healthy from afflicted tissue [2]. In the case of MRI, to overcome this limitation, first, the sequence pulse and free-induction-decay (FID) acquisition are modified to enhance the signal and increase the image resolution [3]. Second, relays on the intravenous injection of high paramagnetic and water-soluble compounds to accelerate $^1$H water nuclei relaxation increases the amount of data per second acquired, which translates into high-resolution images. These compounds, known as contrast agents (CA), are based commonly on iron oxide, lanthanide nanoparticles, and complexes [4]. The family of lanthanide complexes shaped by $Eu^{2+}$, $Eu^{3+}$, $Gd^{3+}$, and $Tb^{3+}$ stand out among other complexes due to their remarkable paramagnetic behavior, which makes them ideal candidates for contrast agents in MRI [5,6].

Although they are extensively employed nowadays, their use is contraindicated in pregnant women, aged individuals, and kidney disease patients due to their toxicity at high doses [7,8].

Lanthanide ions have high toxicity due to their similar ionic radius with $Ca^{2+}$ ions, producing transmetalation reactions with enzymes and proteins, forming gadolinium depositions in the kidneys, and even affecting brain $Ca^{2+}$ modulations. For this reason, the

conception of complexes with high chemical stability and low reactivity, and hence high thermodynamic and kinetic stability, is imperative [9].

Various strategies have been developed to decrease the adverse effects of CA, including increasing solubility to improve the excretion rate, using less-toxic ions such as iron oxide or other transition metal oxides, and, finally, increasing the relaxivity of these compounds [10–14]. In this paper, we focused on the latter, because in this way the dose concentration will be reduced, and the technique resolution will increase, diminishing the risk of collateral effects on human health.

The design of these complexes follows two trends: linear agents such as Magnevist® and Omniscan®, which do not fully surround the lanthanide ion, and macrocyclic agents such as Dotarem® that do this using a cage-like ligand [15]. These latter compounds have lower metal bleeding and therefore fewer side effects, making them preferred for the synthesis of highly stable CA. Once their thermodynamic and kinetic stability are covered, it is possible to focus on the strategies to increase their relaxivity without affecting these requirements [16,17].

Herein, we present new pathways for the development of CA with potentially high relaxivity by applying the Solomon–Bloembergen–Morgan equations within the Redfield limit as an activity predictor for the synthetic strategy [18,19]. We proposed the following two synthetic pathways to obtain highly active MRI contrast agents based on trivalent lanthanides: Huisgen cycloadditions to build organometallic tetramers with cyclen (1,4,7,10–Tetrazacyclododecane) as a building block and 1,4–phenylenediamine functioning like a molecular scaffold; and direct coupling reactions with DOTA as a starting material to produce macromolecular systems active in MRI. We established different parameters impeding the acquisition of the final products, and finally, we proposed different routes to overcome the challenges found during synthesis for future experiments.

*Designing an Efficient Contrast Agent*

CA interaction with the $^1$H protons of water is governed by dipolar interactions; therefore, the distance between the electronic and the nuclear spin defines the interaction strength and the relaxation speed; the shorter the distance, the faster the relaxation. Hence, in a metal ion, the relaxivity will depend on the $^1$H proton interaction with the inner, second, and outer spheres of the lanthanide ion (Equations (1) and (2)) [10].

$$\frac{1}{T_{i,\,p}} = \left(\frac{1}{T_{i,\,p}}\right)^{IS} + \left(\frac{1}{T_{i,\,p}}\right)^{OS} \tag{1}$$

$$r_i = r_i^{IS} + r_i^{2nd} + r_i^{OS},\ i = 1,2 \tag{2}$$

Following the general Solomon–Bloembergen–Morgan (SBM) theory, the inner sphere contribution to the rate relaxation improvement results from the chemical exchange of $^1$H water protons closest to the lanthanide ion with the solution, in which one area is much more populated than the other. The relaxation of these $^1$H protons is governed by dipole–dipole and contact interactions, also called scalars; however, the latter is virtually negligible at frequencies above 10 MHz, since the correlation times of these contributions tend to zero at high frequencies [10]. Therefore, by applying the SBM equations of the paramagnetic relaxation theory, these terms are obtained (Equations (3)–(6)) [20].

$$\left(\frac{1}{T_1}\right)^{IS} = \frac{cq}{55.5}\frac{1}{T_{1m} + \tau_m} = P_m\frac{1}{T_{1m} + \tau_m} \tag{3}$$

$$\left(\frac{1}{T_2}\right)^{IS} = \frac{P_m}{\tau_m}\frac{T_{2m}^{-2} + \tau_m^{-1}T_{2m}^{-1} + \Delta\omega_m^2}{\left(\tau_m^{-1} + T_{2m}^{-1}\right)^2 + \Delta\omega_m^2} \tag{4}$$

$$\frac{1}{\tau_{ci}} = \frac{1}{\tau_R} + \frac{1}{T_{ie}} + \frac{1}{\tau_m}, where\ i = 1,\ 2 \tag{5}$$

$$\frac{1}{\tau_{sci}} = \frac{1}{T_{ie}} + \frac{1}{\tau_m}, where\ i = 1,\ 2 \tag{6}$$

These equations condense the inner sphere relaxivity calculations, where c is the molar concentration of the CA and q is the number of water molecules bonded to the lanthanide center, called the hydration number, and depends on the ion, the structure, and the size of the complex. $\frac{1}{T_{1m}}$ y $\frac{1}{T_{2m}}$ are the relaxation times of the $^1$H water proton closest to the metal ion. The correlation time for the magnetic fluctuation ($\tau_c$) is the shorter term of the rotational correlation time ($\tau_R$), $T_{1e}$ or $T_{2e}$ are the electronic relaxation time, $\tau_m$ is the water residency time, and $\tau_{sci}$ is the scalar correlation time [10].

In Equation (3), it is clear that if the variable $q$ increases, the relaxivity will increase with it. However, a rise in this variable may reduce the thermodynamic stability and kinetic inertness. The $\tau_m$ is one of the most important variables in the SBM equations given that decreasing its value increases the number of molecules relaxed per second. Thus, enhancing the rate of the $^1$H proton's chemical exchange with other water molecules in the bulk improves the relaxivity of the CA. Zero Field Splitting (ZFS) describes the multiple degenerated micro-energy levels of a molecule or ion in the presence of one or more unpaired electrons due to the Zeeman Effect produced by the mutual magnetic dipole-dipole interaction of each unpaired electron. Therefore, using several $Gd^{3+}$ close to each other may enhance the magnetic fluctuations due to the spin density in the complex, improving the $^1$H proton relaxation rate [21].

The theoretical calculations using the SBM equations within the Redfield limit show the relationship between molecular weight and relaxivity. Nevertheless, this augmentation reaches a limit after a few kDa [22]. Although weight is directly related with $\tau_R$, there are more factors involved such as rigidity, which, regarding the macromolecule structure of the CA, can be under fast local or global correlation times ($\tau_i$, $\tau_M$). Following the Lipari and Szabo model, $\tau_i$ affects the expected relaxivity increase (Equations (7) and (8)) [23–25].

$$J(\omega) = \frac{2}{5} \left[ \frac{S^2 \tau_M}{1 + \omega^2 \tau_M^2} + \frac{(1 - S^2) \tau_i}{1 + \omega^2 \tau_i^2} \right] \tag{7}$$

$$\tau_M = \frac{4\pi\eta r_{eff}^3}{3 k_B T} \tag{8}$$

The restriction of the motion causes S to tend to one, causing the $\tau_i$ value to tend to zero and the spectral density function ($J(\omega)$) to depend on $\tau_M$, therefore long correlation times will increase its value and the relaxivity. Hence, the strategy to increase $\tau_M$ besides the molecular weight is to restrict the rotational and translational movements of the CA [26]. For this, different restrictive models were designed for their relaxivity study. Schemes S1–S3 show the compilation of these strategies. It is fundamental to highlight that the increase in $Gd^{3+}$ ions per molecule can enhance their relaxivity values due to the overlapping of their coordination spheres, increasing the interaction area [27]. In this paper, we focused on the design of heterolanthanide complexes to study the relaxation with ions with different total spins, S, and structural values to acquire efficient contrast agents [10,28].

## 2. Materials and Methods

All commercially obtained chemicals were of analytical grade, used without further purification, and supplied by Oxford University in Oxford, UK. $^1$H and $^{13}$C NMR spectra were recorded at 400 MHz using a Bruker Spectrospin DPX-400. All chemical shift ($\delta$) values are given in parts per million. Low-resolution mass spectra were recorded on a Waters Micromass LCT Premier XE spectrometer. Accurate masses were determined to four decimal places using Bruker μTOF and Micromass GCT spectrometers at the Chemistry Research Laboratory of the University of Oxford. Dialysis tubing was performed on a

Dispo-Biodialyzer MWCO 1 kDa and Cuprisorb® as the copper-adsorbing resin. HPLC purification was acquired with the Waters HPLC or a UV-Vis spectrometer (Cary 60 UV–Vis, Agilent Technologies, Santa Clara, CA, USA). Elemental analysis for H, C, and N was carried out on a Perkin Elmer 2400 Series II CHNS Analyzer. Metal ions were analyzed on an Inductively Coupled Plasma Mass Spectrometry on an Agilent 7700 ICP–MS instrument.

The family of complexes in this study were the heterolanthanide compounds based on azide–alkyne Huisgen cycloadditions built like a potential CA: a tetramer with a certain rotational impediment, an open trimer, and another closed. The first one was designed using DOTA units, a 1,2,3–triazole ring, and 1,4–phenylenediamine (Figure 1). The second was a trimer $Gd^{3+}$–$Tb^{3+}$ complex with an open configuration to measure the relaxivity in CA–hindered and a tetramer in a closed configuration with high rigidity (Figures 2 and 3).

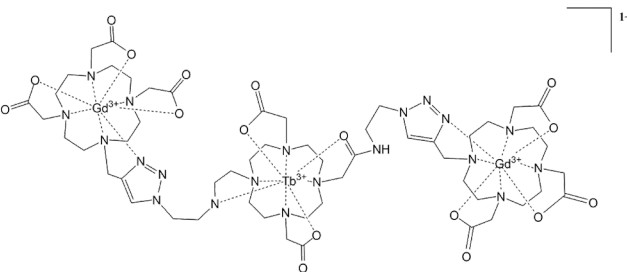

**Figure 1.** Open trimeric heterolanthanide complex.

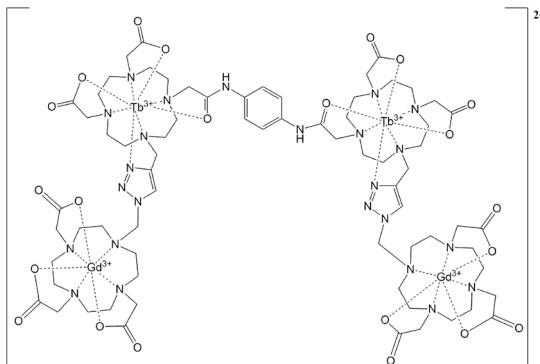

**Figure 2.** Open tetrameric heterolanthanide complex.

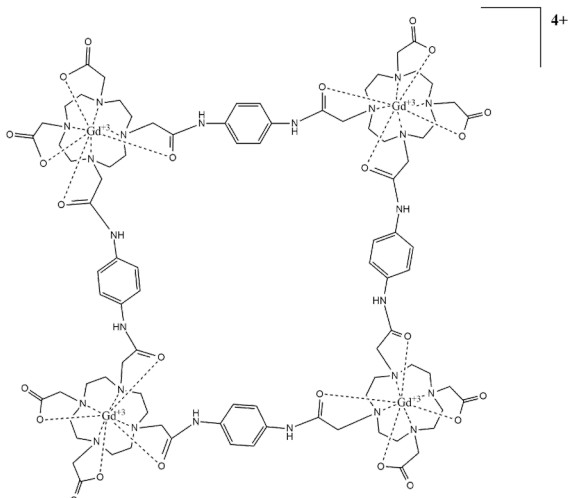

**Figure 3.** Closed tetrameric lanthanide complex.

## 3. Results

The synthetic pathway for each of the following molecules with their supported spectroscopic characterization can be found in Figures 4 and 5 and in the Supplementary Information [29–34].

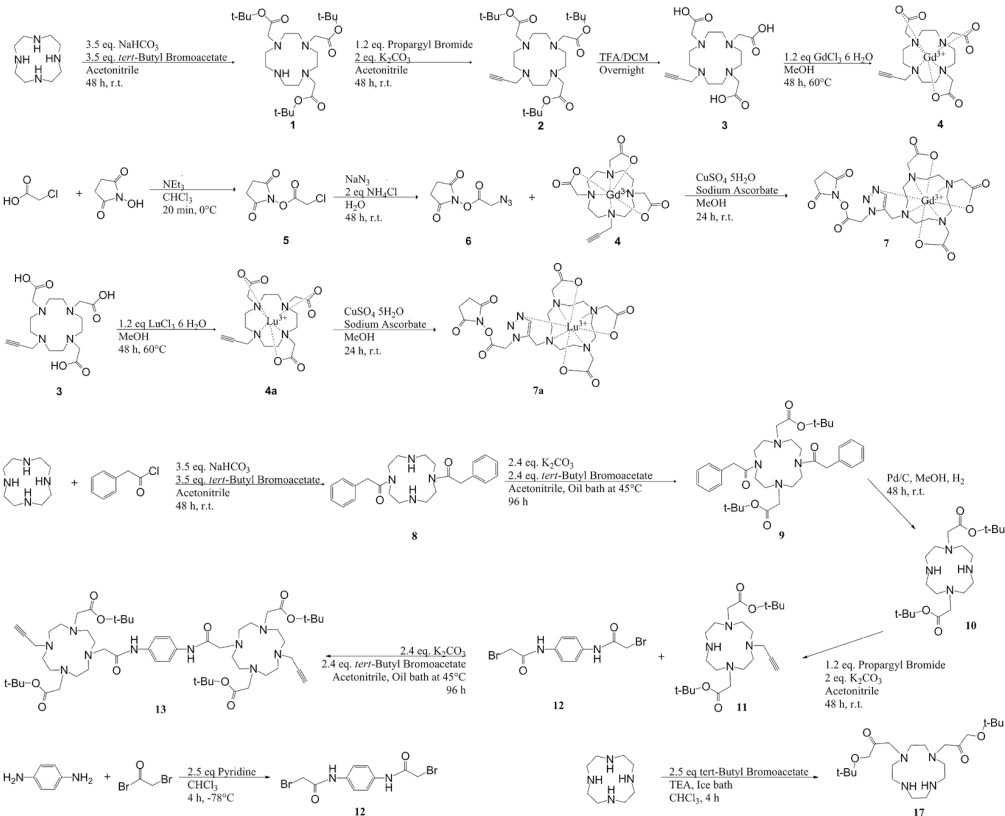

**Figure 4.** Synthetic pathway for molecules 1–13 and 17.

**Figure 5.** Synthetic pathway for molecules 14–18.

The different synthetic routes used for the synthesis of compound 18 are shown in Table 1. Every single reaction was performed in duplicate, purified by flash column chromatography (FCC) and preparative TLC using as eluents DCM (90%):MeOH (9.9%):TEA (0.1%) for starting materials and toluene (30%):hexane (69.9%):TEA (0.1%) for the expected product, resulting in minimal amounts of product, enough for the $^1$H–NMR and MS studies.

**Table 1.** The reaction between compounds 12 and 17 is a possible way to reach CisTetraDOTA. The NMR and MS results suggest that the reaction produced an opened and closed trimer. The evidence was elusive (spectrum 18 and chromatogram 7); even with HPLC and TLC preparative purifications, it was not possible to remove the presence of starting materials on both results, affecting the identification of the product of these reactions. However, this result could lead to a new kind of synthetic route to make a new closed CA, thus, this finding will be studied in future projects. Not successful: -; undefined: *.

| Solvent | Base | Base Eq. | T (°C) | Time (h) | Eq. C17 | Eq. C12 | Result | Yield (%) |
|---|---|---|---|---|---|---|---|---|
| **Chloroform** | TEA | 3 | r.t. | 3 | 1 | 5 | - | - |
| **Chloroform** | Pyridine | 3 | 0 | 12 | 1 | 1 | - | - |
| **Chloroform** | Pyridine | 5 | −78 | 12 | 1 | 1 | - | - |
| **DMF** | TEA | 10 | r.t | 12 | 1 | 5 | - | - |
| **DMF** | TEA | 10 | r.t | 12 | 1 | 2 | - | - |
| **DMF** | TEA | 10 | 50 | 12 | 1 | 1 | - | - |
| **DMF** | $Cs_2CO_3$ | 5 | 50 | 12 | 1 | 2.2 | - | - |
| **DMF** | $Cs_2CO_3$ | 10 | 60 | 12 | 1 | 2.2 | - | - |
| **Toluene** | $Cs_2CO_3$ | 10 | 110 | 12 | 1 | 2.2 | - | - |
| **Toluene** | $Cs_2CO_3$/1% NaI | 10 | 110 | 12 | 1 | 2.2 | - | - |
| **Toluene** | $Cs_2CO_3$/5% NaI | 10 | 110 | 24 | 1 | 2.2 | - | - |
| **Toluene** | $Cs_2CO_3$/5% NaI | 10 | 110 | 48 | 1 | 2.2 | - | - |
| **Acetonitrile:0.1 DMF** | $Cs_2CO_3$/5% NaI | 10 | 110 | 48 | 1 | 2.2 | * | * |
| **Acetonitrile** | $Cs_2CO_3$/5% NaI | 10 | 82 | 24 | 1 | 2.2 | - | - |
| **Acetonitrile** | $NaHCO_3$ | 2 | 82 | 48 | 1 | 2.2 | - | - |
| **Acetonitrile** | $NaHCO_3$ | 2 | 82 | 56 | 1 | 2.2 | - | - |
| **DMF** | TEA | 5 | 80 | 48 | 1 | 2.2 | - | - |
| **DMF** | $NaHCO_3$ | 5 | 80 | 48 | 1 | 2.2 | - | - |
| **Toluene:0.1 DMF** | $Cs_2CO_3$/5% NaI | 10 | 110 | 48 | 1 | 2.2 | * | * |

## 4. Discussion

### 4.1. Click Chemistry Synthetic Pathway

Our first approach to the synthesis of highly active CA for MRI was the use of the extensively known azide–alkyne Huisgen cycloaddition protocol because it offers a reliable and selective strategy for the rapid synthesis of adducts through heteroatom links. Our goal was to synthesize a linear heterolanthanide ($Gd^{3+}$–$Tb^{3+}$) tetramer using DOTA-derivative blocks and 1,4–phenylenediamine as a rigid molecular scaffold.

The first complex synthesized (compound 7, Figure 4) was composed of a DOTA derivative with an N–Hydroxysuccinamide (NHS) terminal to produce an adduct of high molecular weight by direct coupling with peptides and proteins. The compound was successfully synthesized (73.2%, 0.123 mmol) and characterized using $Lu^{3+}$ as an ion to check the NMR shifts due to its diamagnetic behavior and similar ionic radius to $Gd^{3+}$. The characterization of this compound can be found in the Supplementary Information.

Two significant factors were found at the end of this synthesis: first, the NHS ester had a low lifetime in the complex due to the quick hydrolysis with water, and second, the $Cu^{+}$ was retained by the 1,2,3–triazole ring with traces of 0.63–0.83 ppm. Both effects decreased their functionality and jeopardized their use in humans due to the high toxicity of Cu(I) and losing the NHS terminal diminished the possibility of making adducts between the potential CA with biomolecules of high molecular weight such as proteins.

To tackle the limiting factors impacting the reaction, we first produced an in situ NHS ester synthesis without further purification, which allowed the addition of nucleophiles directly and the formation of the final adduct [35]. Regarding the $Cu^{+}$ presence, it was persistent after several dialysis processes and filtration using CupriSorb$^{TM}$, losing 10–13% of the product at each step. In addition, the content of $Cu^{+}$ in these synthons was present in both the $Gd^{3+}$ complexes and the ligands. This absorption could be due to the closeness of folding the triazole ring arm with the nitrogen on the tetraazadodecane amines and the carbonyl of the carboxylic acids and esters. This unexpected result hampered the synthesis of our proposed macromolecule family; however, the strategy for the design of these new synthon macro-complexes can be used for the development of heterolanthanide arrays ($Tb^{3+}$ and $Eu^{3+}$) for molecular imaging by modifying the tert-butoxide arms to add fluorescent probes for lanthanide luminescence via the transfer of fostered resonance energy [36–38].

These results forced the project to move forward in the search for new synthons to prove our strategy. The next step was setting the same parameters but without the addition of any ion of a transition metal catalyst to avoid another unwanted complex. This was the CisTetraDOTA synthesis.

### 4.2. cisTetraDOTA Synthetic Pathway

Our final study was the direct synthesis of a macromolecular contrast agent using DOTA as a building block. This final complex was designed using the crowding effect, reducing the distances via a rigid linker to increase rotational movement while limiting translational movement by steric hindrance [39]. This will affect the rotational correlation $\tau_R$ and the hydration number in the second sphere and outer sphere, due to the interception of these coordination spheres of each complex in the macromolecule. Besides the high molecular weight, this will balance the $\tau_R$ of the outer sphere since the reorientational correlation time divides into local and global movements. According to the free model of Lipari–Szabo, if both movements are sluggish, the macro complex will have better relaxivity [10,40–42].

The synthesis of this compound was planned over several strategies, with compound 12 as a molecular scaffold and the Cis-DO2A, compound 17, as the tetramer's body. The different synthetic routes used are shown in Table 1. Every single reaction was performed in duplicate and purified by Flash Column Chromatography (FCC) and preparative TLC. In order to purify the samples by FCC and TLC, two solutions were used as eluents. To obtain a low RF for unreacted starting materials with a high relative polarity, polar eluents (DCM (90%):MeOH (9.9%):TEA (0.1%)) were used, while nonpolar eluents (toluene (30%):hexane (69.9%):TEA (0.1%)) were used to separate multiple products with significantly lower polarity after each reaction.

This candidate had a higher theoretical relaxivity due to its high molecular weight, rigidity, and multiple $Gd^{3+}$ ions, which increase the relaxivity on the inner, second, and outer spheres, as was detailed through Equations (1)–(9). In addition, the higher anisotropy and symmetry provided by the N,N′–(1,4–phenylene) diacetamide bridge make it suitable

for optical image complexes based on $Eu^{3+}$ and $Tb^{3+}$. Nonetheless, the results were inconclusive given the limited time available to fully isolate and characterize the proposed molecule. The multiple efforts and alternative pathways are listed in Table 1.

Self-assembling behavior in this compound was expected through a double N-alkylation over compound 17 to produce cisTetraDOTA, but the evidence provided by the $^1$H-NMR and mass spectrometry did not support this hypothesis (spectrum S18 and chromatogram S7, Supplementary Information). On the other hand, these results showed initial evidence of a possible trimer with a molecular weight of 1765.09 Da with an m/z on chromatogram S7 of 1718.220 Da (3.92%), corresponding to the possible $\alpha$-cleavage of one tert-butoxy radical (73.07) from the esters with one Na+ ion $(C_{86}H_{135}NaN_{18}O_{17})^+$. A second pattern was found on 1282.823 Da (22.62%), probably belonging to the $\alpha$-cleavage of the 66.7% of the tertbutyl esters and one N,N'–(1,4–phenylene) diacetamide bridge $(C_{64}H_{98}N_{16}O_{126})^+$. The final m/z peak placed on 574.409 Da (22.41%) was assigned to the cisDO2A bonded to one N,N'–(1,4–phenylene) diacetamide, both with an $\alpha$-cleavage fragmentation $(C_{29}H_{47}N_6O_6)^+$. However, at 1848.191 Da (1.68%), the chromatogram showed what seemed to be an open trimer with the bromide still attached to the N,N'–(1,4–phenylene) diacetamide (1845.02 Da) $(C_{90}H_{145}BrN_{18}O_{18})^+$. Chromatogram S8 showed more evidence about this reaction and the possible compound obtained. At 1801.2 m/z, it was possible to see the molecular ion peak, which fit with the dihydrate trimer $(C_{90}H_{144}N_{18}O_{18}{}^{\bullet+}\cdot 2\ H_2O$, Figure S1). The second most abundant fragment was the peak of 1137.7 m/z belonging to the dimer $(C_{58}H_{96}N_{12}O_{11}{}^{\bullet+})$, and finally the base ion peak at 589.4 m/z belonged to the monomer fragment $(C_{30}H_{49}N_6O_6{}^+)$.

In order to obtain a clear picture of the experimental NMR, some DFT simulations were carried out in the present work at the B3LYP/cc–PVDZ level using the basis set function in the Gaussian 09 package [43]. Molecular optimizations and NMR shifts were computed in the gaseous phase and also by applying a gauge-invariant atomic orbital (GIAO) approach for the NMR [44]. The input structure was modified by the tert–butyl groups exclusion to save computational time. Two different structures were considered: an open and close macrocycle trimer (Figure 6).

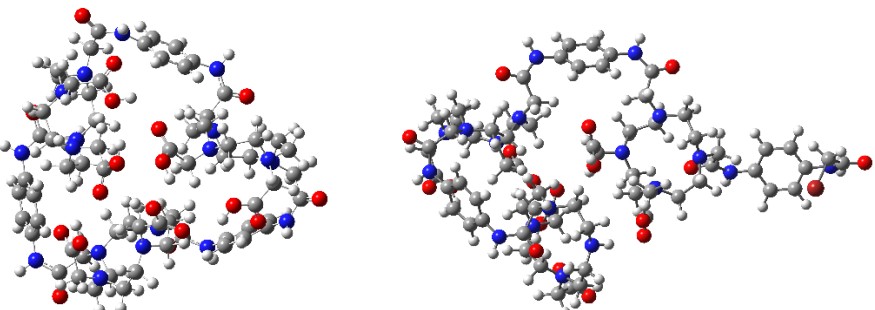

**Figure 6.** Optimized structures of the two considered system. Closed trimer, (**left**) and open trimer, (**right**) (C = gray, O = red, H = white and N = blue).

The theoretical $^1$H chemical shifts were calculated for the optimized geometries; the results showed that the range of the $^1$H chemical shifts of the typical organic molecules were usually 100 ppm. [45–47]. The accuracy ensured the reliable interpretation of the spectroscopic parameters. Spectra S19–S24 are available in the Supplementary Information. Two significant peaks were analyzed among the experimental (400 MHz Bruker NMR) and predicted $^1$H–NMR results; the shift at δ 3.50 ppm (d, 24H) belonging to the $\alpha$–carbons bonded to the tertiary macrocycle amine, which showed that equivalent protons were the evidence of a symmetrical adduct, and the shift at δ 3.90 ppm (s, 2H) pertaining to the existence of a single methylene bromide ($CH_2$–Br). Both signals had a clear difference in the possible outcomes: the $\alpha$-carbons for the open trimer predicted $^1$H–NMR reported a wide signal distribution between 3.0–5.0 ppm (S20 spectrum), an expected output due to the low symmetry. On the other hand, the closed trimer showed a less pronounced peak distribution

between 3.5–3.8 ppm (S23 spectrum), similar to the one obtained experimentally. The second signal was relevant to this study because its presence denoted an incomplete ring formation. Protons H–129 and H–130 belonged to methylene bromide in the open predicted [1]H–NMR with chemical shifts at 4.3 ppm and 4.85 ppm (S20 Spectrum); one of these signals was found at 4.65 ppm in the experimental spectra integrating with two protons (S18 spectrum). The presence of both signals in the experimental [1]H–NMR supported by the predicted [1]H–NMR showed the incomplete reaction between compounds 11 and 17 with the formation of an open and closed trimer, detected by NMR and MS. Despite the efforts in the purification step, both products could not be effectively isolated (Figure 7); however, further analysis was not possible due to impurities and the minimal amount of product obtained.

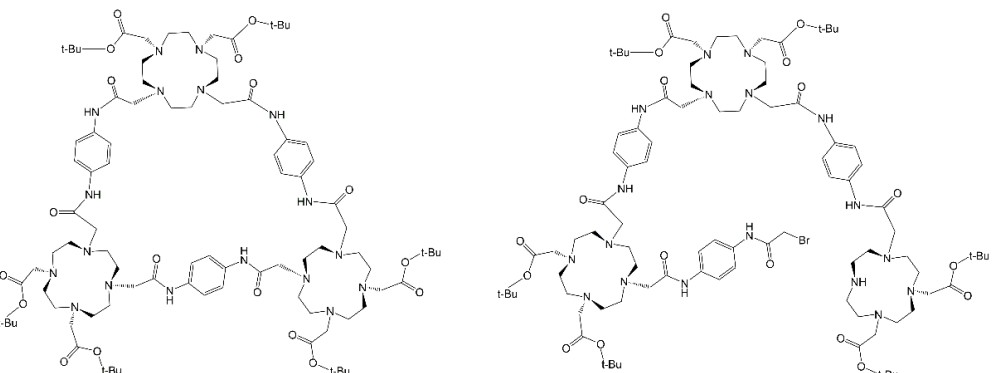

**Figure 7.** Potential synthons obtained instead of G4–CisTetraDOTA following the spectral information.

The low yield of this reaction can be explained through the partial solubility of compound 17 in organic solvents, decreasing the effective collisions with compound 12. For that reason, several solvents were used with different polarities, i.e., protic and non–protic, mixed and pure, but it did not affect the course of the reaction.

DFT calculations showed that the sp3 HOMO–LUMO from the secondary amines of compound 17 had a different orientation, with one of the HOMOs looking inside the tetraazacyclododecane ring (Figure 8 atom 15). This out-of-phase HOMO decreased the number of effective collisions in the $S_N2$ reaction with the N,N′–(1,4–phenylene) diacetamide dibromide, besides increasing the steric hindrance for the N–alkylation.

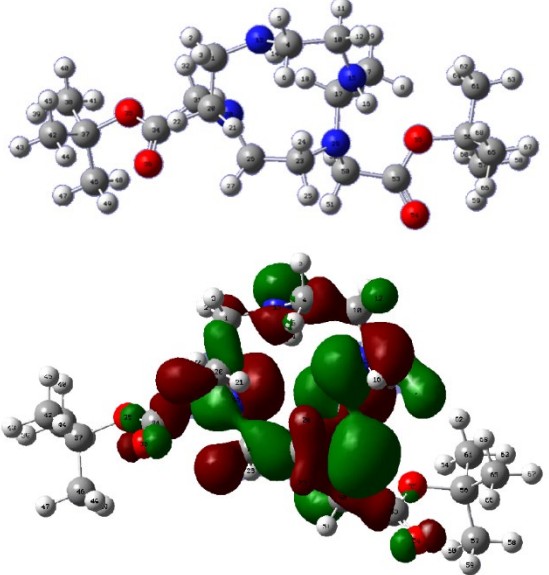

**Figure 8.** DFT calculations compound 17.

The same calculations using different solvents, such as protic and aprotic, revealed that the molecular orbital from secondary amines did not change when the nature of the solvents changed. These calculations explained the results of how several N–alkylation protocols were tried with no positive results. The use of different temperatures and long exposure did not affect the course of the reaction. This means that the low yield of the double N–alkylation between cisDO2A and the N,N′–(1,4–phenylene)diacetamide dibromide scaffold had less chance of completion due to steric hindrance, out-of-phase HOMO, and the partial solubility of compound 17. However, Jordan D. Goodreid, Petar A. Duspara, Caroline Bosch, and Robert A. Batey, [48] developed a direct coupling of primary amines with lithium carboxylates, which could increase the chances of producing cis–etraDOTA. In addition, the implementation of copper-free click reactions with DBCO and HBTU and EDC coupling protocols could improve the chances of acquiring some of the proposed molecules [49–52].

Despite the fact that the final complex was hampered by the aforementioned limitations, we presented effective synthesis pathways to produce synthons with foreseeable applications on tunable and luminescent probes as lanthanide-metal–organic frameworks [53,54]. These results showed that, beyond theoretical activity, achieving the total synthesis of such macromolecules requires several steps where toxic and persistent contaminations can appear. Our next objective is based on designing the best synthetic route to obtain a new family of compounds, emphasizing the potential trimer obtained, which, if synthesized with high yields, will stand out not only as a CA for MRI but also as a superb paraCEST agent with Tb(III) or Dy(III) ions and a scaffold for lanthanide-based metal–organic frameworks and MRI-fluorescent probes due to its rigid structure and expected high rotational correlation time [55].

## 5. Conclusions

Herein, we have put in place a strategy to design Cas with high activity based on the Solomon–Bloembergen–Morgan theory. Two main molecular probes arose from the calculations: a closed trimer and two building blocks for click chemistry. Following this, the synthetic methodology attempted to encounter various difficult steps that hampered the attainment of a contamination-free final complex. Computational calculations gave relevant insight into the main factors limiting the reactivity of compound 12 or cisDO2A through $S_N2$ reactions.

The strong interaction between the products and silica (FCC and TLC) negatively affected the purification steps, resulting in a significant decrease in product recovered and, therefore, in yields. In order to overcome this, we found that DFT calculations could provide precise information about the relative polarity and, therefore, determine the optimal solvents and anti-solvents for liquid–solid or liquid–liquid extractions, recrystallization, and trituration.

Given that the contaminations represented a significant risk for their use as synthons for CA, alternative routes were proposed, including the implementation of copper-free click reactions with DBCO, HBTU, and EDC. Nonetheless, the resulting synthons were foreseeably good building blocks for the production of efficient heterolanthanide luminescent probes or paraCEST agents. Overall, we hope that this work lays the groundwork for a comprehensive route to design synthons and complexes with applications in luminescent probes and high-relaxation MRI contrast agents.

**Supplementary Materials:** The following supporting information can be downloaded at: https://www.mdpi.com/article/10.3390/app12199969/s1. Scheme S1: Triester, Compound 1; Scheme S2: Propargyl Triester, Compound 2; Scheme S3: Propargyl DO3A, Compound 3; Scheme S4: Lutetium-Propargyl DO3A, Compound 4; Scheme S5: Compound 5; Scheme S6: Compound 6; Scheme S7: TMo1, Compound 7; Scheme S8: Compound 8; Scheme S9: Compound 9; Scheme S10: Compound 10; Scheme S11: Compound 11; Scheme S12: Compound 12; Scheme S13: Compound 13; Scheme S14: Compound 14; Scheme S15: Compound 15; Scheme S16: Compound 15; Scheme S17: Compound 16; Scheme S18: Possible compound obtained; Scheme S19: Predicted 1H-NMR open trimer; Scheme

S20: Predicted [1]H-NMR open trimer, range 2–5 ppm; Scheme S21: Predicted [1]H-NMR open trimer, range 6–9 ppm; Scheme S22: Predicted [1]H-NMR closed trimer; Scheme S23: Predicted [1]H-NMR closed trimer, range 2–5 ppm; Scheme S24: Predicted 1H-NMR closed trimer, range 7–10 ppm; Chromatogram S1. Compound 7; Chromatogram S2. Compound 10; Chromatogram S3. Compound 12; Chromatogram S4. Compound 13; Chromatogram S5. Compound 16; Chromatogram S6. Compound 17; Chromatogram S7. Possible compound obtained; Chromatogram S8. Possible compound obtained; Figure S1: Mass fragments Chromatogram S8 Compound 18.

**Author Contributions:** Conceptualization, C.G.; methodology, C.G.; software, R.S.-M.; validation, J.U.; formal analysis, C.G.; investigation, C.G.; resources, R.S.-M. and J.U.; data curation, J.U. and R.S.-M.; writing—original draft preparation, C.G.; writing—review and editing, C.G. and J.U.; visualization, J.U.; supervision, J.U.; project administration, J.U.; funding acquisition, J.U. All authors have read and agreed to the published version of the manuscript.

**Funding:** This research was funded by Departamento Administrativo de Ciencia, Tecnología e Innovación (COLCIENCIAS), grant number 727-2015 and the APC was funded by Banco de la República de Colombia project 4.451.

**Institutional Review Board Statement:** Not applicable.

**Informed Consent Statement:** Not applicable.

**Data Availability Statement:** Not applicable.

**Acknowledgments:** This project would not have been possible without the advice, encouragement, materials used for experiments, and technical support provided by Professor Stephen Faulkner and the Chemistry Department at University of Oxford.

**Conflicts of Interest:** The authors declare no conflict of interest.

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
