# Peer review of "New Strategies for Potential Contrast Agents’ Synthons Highly Active to MRI Based on Gd3+, Eu3+, and Tb3+"

_applsci, doi:10.3390/app12199969_

Round 1

Reviewer 1 Report

The presented document shows the synthesis of molecules that could be used in the diagnosis of diseases by imaging. This field has always required new molecules with more appropriate properties for imaging.

The most interesting of the work are the synthesis routes of new compounds derived from Eu, Gd, and Tb. However, the authors show that the synthesis of the designed complexes is not efficient enough, given that they obtained minimal amounts of product, just enough to make the chemical characterization of complexes by NMR and mass analysis.

The document shows a wide scientific value due to the type of synthesized molecules, but it has a great weakness in not defining a route with high efficiency to obtain the designed molecules, in addition to not being able to define an efficient purification process.

The purpose of the article is excellent, although apart from the experiment it is not entirely adequate, not because it is out of context, but because it does not produce the best results, that is, it shows high efficiency in the synthesis of molecules for image diagnosis.

I consider that the article can be published because it shows an excellent foundation of the experimental and methodological part, which can be improved through deeper experimentation.

The references used are completely adequate, and to a sufficient degree, and correspond to recent works, which adequately supported the results obtained.

Author Response

Dear Dr. Reviewer,

We want to extend our gratitude for your comments. The synthesis of these challenging complexes demands plasticity and recursiveness. We did our best, and we are proud of our research; however, COVID-19 negatively affected our chances to go further and achieve the best results that we were projecting. This research also exhibits our commitment to contributing to this amazing field, and we hope soon we can provide new insights into the design of highly efficient contrast agents.

Thanks for being part of the help that we needed to make a robust research paper.

Reviewer 2 Report

The authors present alternative synthetic strategies for the development of theoretically high relaxivity synthons based on lanthanides using the Solomon – Bloembergen – Morgan equations through click-chemistry and direct addition. Their preliminary results showed that both the mathematical background and synthetic strategy needed to be revised.

 Overall, the findings are interesting but some minor concerns are needed to address before the final publication of the manuscript as given below.

1.     There are numerous typographic errors in the manuscript which are needed to be corrected.

2.     Lanthanides are rare earth elements then why did the author select the expensive metals? Instead of it many transition metals can perform a better role and are also very cheap and abundantly available with reduced side effects.

3.     What kind of solvents are used in Column chromatography and TLC?

4.     Although the work is very impressive still the purity and yield are major problems. What strategies will be adapted to overcome these problems?

5.     The Authors must cite some new recent references from the year 2022.

6.     It is suggested to perform elemental analysis to further support the synthesis of compounds

Author Response

Dear Dr. Reviewer,

We want to extend our gratitude for your valuable comments and the short-time response and for leading us to obtain a robust research paper. 

Best regards,

Carlos Guzmán, Rubén Soria-Martínez and Julián Urresta
